# Dietary quality and cardiometabolic indicators in the USA: A comparison of the Planetary Health Diet Index, Healthy Eating Index-2015, and Dietary Approaches to Stop Hypertension

Sarah M. Frank[1,2], Lindsay M. Jaacks[2], Christy L. Avery[1,3], Linda S. Adair[1,4], Katie Meyer[4,5], Donald Rose[6], Lindsey Smith Taillie[1,4]*

1 Carolina Population Center, University of North Carolina at Chapel Hill, Chapel Hill, North Carolina, United States of America, 2 Global Academy of Agriculture and Food Systems, University of Edinburgh, Midlothian, United Kingdom, 3 Department of Epidemiology, Gillings School of Global Public Health, University of North Carolina at Chapel Hill, Chapel Hill, North Carolina, United States of America, 4 Department of Nutrition, Gillings School of Global Public Health, University of North Carolina at Chapel Hill, Chapel Hill, North Carolina, United States of America, 5 Nutrition Research Institute, University of North Carolina at Chapel Hill, Kannapolis, North Carolina, United States of America, 6 Tulane Nutrition, School of Public Health and Tropical Medicine, Tulane University, New Orleans, Los Angeles, United States of America

* taillie@unc.edu

## Abstract

### Background

The Planetary Health Diet Index (PHDI) measures adherence to the sustainable dietary guidance proposed by the EAT-*Lancet* Commission on Food, Planet, Health. To justify incorporating sustainable dietary guidance such as the PHDI in the US, the index needs to be compared to health-focused dietary recommendations already in use. The objectives of this study were to compare the how the Planetary Health Diet Index (PHDI), the Healthy Eating Index-2015 (HEI-2015) and Dietary Approaches to Stop Hypertension (DASH) relate to cardiometabolic risk factors.

### Methods and findings

Participants from the National Health and Nutrition Examination Survey (2015–2018) were assigned a score for each dietary index. We examined disparities in dietary quality for each index. We used linear and logistic regression to assess the association of standardized dietary index values with waist circumference, blood pressure, HDL-C, fasting plasma glucose (FPG) and triglycerides (TG). We also dichotomized the cardiometabolic indicators using the cutoffs for the Metabolic Syndrome and used logistic regression to assess the relationship of the standardized dietary index values with binary cardiometabolic risk factors. We observed diet quality disparities for populations that were Black, Hispanic, low-income, and low-education. Higher diet quality was associated with improved continuous and binary cardiometabolic risk factors, although higher PHDI was not associated with high FPG and was the only index associated with lower TG. These patterns remained consistent in sensitivity analyses.

**Data Availability Statement:** The data underlying the results presented in the study are available from the National Health and Nutrition Examination Survey website, https://www.cdc.gov/nchs/nhanes/index.htm.

**Funding:** SMF, LMJ, and LST received funding from Wellcome Trust Award Number 216042/Z/19/Z, https://https://wellcome.org/ The funders had no role in study design, data collection and analysis, decision to publish, or preparation of the manuscript.

**Competing interests:** The authors have declared that no competing interests exist.

## Conclusions

Sustainability-focused dietary recommendations such as the PHDI have similar cross-sectional associations with cardiometabolic risk as HEI-2015 or DASH. Health-focused dietary guidelines such as the forthcoming 2025–2030 Dietary Guidelines for Americans can consider the environmental impact of diet and still promote cardiometabolic health.

## Introduction

Cardiovascular disease (CVD) is the number one cause of morbidity [1] and mortality [2] in the US. Poor dietary quality, in turn, is the number one risk factor for CVD [1]. Thus, improvements in dietary quality could significantly lessen the burden of CVD in the US.

Dietary guidelines are a set of recommendations designed to promote health and are often used as the basis for food policies. In 2019, the EAT-*Lancet* Commission on Food, Planet, Health introduced a "universal healthy reference diet," [3] to jointly address diet-related disease and the environmental impact of food production. The diet emphasizes one rich in plant-sourced foods and low in animal-sourced foods using suggested amounts for a diet of 2500 kilocalories per day.

The Planetary Health Diet Index (PHDI) is a relatively new measure of dietary quality that incorporates recommendations on and is innovative in its consideration of sustainability and health from the EAT-*Lancet* reference diet into a numerical index [4–7]. To justify incorporating the EAT-*Lancet* Commission's climate-focused recommendations into US food policies, there is a need to assess the PHDI's performance as a predictor of cardiometabolic health and see how it compares to dietary recommendations already in use. Two commonly used dietary indices in the US are the Healthy Eating Index-2015 (HEI-2015) and an index based on Dietary Approaches to Stop Hypertension (DASH). Like PHDI, HEI-2015 uses pre-defined thresholds to quantify adherence to the Dietary Guidelines for Americans (DGAs) but does not discourage animal-sourced foods [8]. DASH is designed to prevent and control hypertension, but unlike PHDI and HEI-2015, DASH is scored on the distribution of component intake within the target population [9]. Both HEI-2015 and the DASH index are associated with decreased risk of cardiometabolic morbidity and mortality in the US [10, 11].

Additionally, there are well-documented dietary disparities by sex, income, education, and race/ethnicity for both HEI-2015 [12] and DASH [13]. To our knowledge, there have been no analyses of disparities in PHDI in the US. There is therefore a need to quantify the disparities in dietary quality as measured by PHDI and compare to disparities in HEI-2015 and DASH.

The objectives of this study were to see how the PHDI correlates with HEI-2015 and DASH. compare the performance of the three dietary indices in terms of prediction of binary cardiometabolic risk factors. We further examine socioeconomic disparities in diet quality as measured by the three indices.

## Materials and methods

### Study population

The US National Health and Nutrition Examination Survey (NHANES) is a repeated cross-sectional survey that uses a multistage probability design to sample the civilian, non-institutionalized population residing in the 50 states and District of Columbia [14]. Two cycles of NHANES are required to obtain reliable estimates of population-level means [15, 16], so we included data from the two most recently available NHANES cycles unaffected by the

COVID-19 pandemic. The study protocols of the NHANES are approved by the Research Ethics Review Board at the National Center for Health Statistics (NCHS) [14]. This is a retrospective study of data that were fully-anonymized before the authors accessed them. Because the de-identified observational data from the National Health and Nutrition Examination Survey are publicly available for download, this study received a determination of Not Human Subjects Research by the Institutional Review Board at [First Author's Home University].

Eligible participants were non-pregnant or lactating individuals aged 20 years or older who participated in the 2015–2016 or 2017–2018 NHANES cycle and for whom two days of valid dietary intake data were available. Participants whose mean total energy intake was <500kcal or >8000kcal/day were excluded [17].

## Assessment of dietary intake

Trained interviewers used the US Department of Agriculture Automated Multiple Pass Method to gather 24-hour dietary recall data [18]. Participants were asked to recall all foods and beverages they consumed the previous day. Measuring guides were used to assist with estimating portion sizes. The second dietary interview was conducted unannounced via phone 3–10 days after the initial face-to-face interview.

Dietary recall data were merged to the Food Patterns Equivalent Database (FPED), which assigns foods to the 37 USDA Food Pattern Components using a food composition table. For single-ingredient food items, FPED assigns foods directly to the corresponding component. For foods with ingredients from more than one component, FPED disaggregates these items into their component ingredients' gram weights using standard recipe files [19].

Dietary recall data were also used to derive total energy intake [20].

## Planetary Health Diet Index, PHDI

The Planetary Health Diet Index (PHDI) measures adherence to the recommendations of the EAT-*Lancet* Commission Scientific Report [3] and is designed to provide 2500 kilocalories/day. The index consists of 14 equally-weighted components worth 10 points each (Table 1, S1 Table). Six of these components (whole grains; whole fruits; non-starchy vegetables; nuts and seeds; legumes; and unsaturated oils) were encouraged and eight (starchy vegetables; dairy; red and processed meat; poultry; eggs; fish; saturated oils and *trans* fats; added sugar and fruit juice) were discouraged. The theoretical range of the PHDI is 0 to 140, with a higher score indicating better adherence.

## Healthy Eating Index, HEI-2015

The Healthy Eating Index (HEI-2015) is a quantitative measure of adherence to the US DGAs, which are dietary recommendations published by the federal government and used as the basis for federal food and nutrition policy [21]. HEI-2015 was calculated based on scores for 13 food components (Table 1): nine adequacy components, for which intake was encouraged (total fruits including fruit juice; whole fruits; total vegetables; greens and beans; dairy; total protein foods; seafood and plant proteins; and ratio of unsaturated: saturated fatty acids) and four moderation components for which intake was discouraged (refined grains; sodium; added sugars; and saturated fats). Participant intakes for each food group were scored based on energy-adjusted food intake (amount per 1000 kilocalories). The minimum and maximum scoring criteria for each food group are described in detail elsewhere, and participant intakes between the minimum and maximum were scored proportionately [22, 23]. Unlike PHDI and DASH, these components are not weighted equally, with seven components (whole grains; dairy; ratio of unsaturated: saturated fatty acids; refined grains; sodium; added sugars; saturated fats)

**Table 1. Comparison of the dietary components of the Planetary Health Diet Index (PHDI), Healthy Eating Index-2015 (HEI-2015) and Dietary Approaches to Stop Hypertension (DASH).**

| Dietary Components | PHDI* | HEI-2015* | DASH‡ |
|---|---|---|---|
| **Encouraged components** | | | |
| Grains | Whole grains | Whole grains | Whole grains |
| Fruits | Whole fruit *(excluding juice)* | Whole fruit† *(excluding juice)* | Total fruit *(including juice)* |
| | | Total fruit† *(including juice)* | |
| Vegetables | Vegetables *(excluding starchy)* | Total vegetables† | Total vegetables |
| | | Greens and beans† | |
| Proteins | Nuts | Total protein foods† | Total nuts and legumes |
| | Legumes | Seafood and plant proteins† | |
| Dairy | | Total dairy | Low-fat dairy |
| Fats & oils | Unsaturated oils | Fatty acids (PUFAs + MUFAS)/ SFAs | |
| **Discouraged components** | | | |
| Grains | | Refined grains | |
| Vegetables | Starchy vegetables | | |
| Proteins | Red/processed meat | | Red/processed meat |
| | Poultry | | |
| | Eggs | | |
| | Fish | | |
| Dairy | Total dairy | | |
| Fats & oils | Saturated oils and *trans* fat | Saturated fats | |
| Sugar | Added sugar and fruit juice | Added sugars (excludes fruit juice) | Sugar-sweetened beverages |
| Sodium | | Sodium | Sodium |

* All dietary pattern component scores range 0–10 unless otherwise noted

† Component score range: 0–5

‡ All component score range: 1–5

assigned a range of 0–10 points, and six components (total fruits; whole fruits; total vegetables; greens and beans; total protein foods; seafood and plant proteins) assigned a range of 0–5 points. Scores were then summed to create the total score (theoretical range: 0 to 100, with a higher score indicating better adherence) [8].

### Dietary Approaches to Stop Hypertension, DASH

Dietary Approaches to Stop Hypertension (DASH) is specifically designed to maintain a healthy blood pressure and has been adapted in settings throughout the globe. The scoring criteria for DASH is based on a total of eight categories (Table 1), five of which were encouraged (fruits; vegetables; whole grains; nuts and legumes; and low-fat dairy) and three of which were discouraged (sodium; sugar-sweetened beverages; and red and processed meat). Scores for each category were assigned by quintile of energy-adjusted food group intake. DASH scores can range from 8 to 40, with a higher score indicating better adherence [11, 23].

### Cardiometabolic risk factors

We examined the cardiometabolic risk factors that are used as the constituent criteria for the clinical definition of Metabolic Syndrome [24]. These cardiometabolic risk factors were: high waist circumference, high blood pressure, reduced high-density lipoprotein cholesterol (HDL-C), high fasting plasma glucose, and elevated fasting triglycerides.

**Table 2. Criteria used to define binary cardiometabolic risk factor outcomes.**

| Cardiometabolic Risk Factor | Threshold |
| --- | --- |
| High waist circumference | ≥102 centimeters in males |
| | ≥88 centimeters in females |
| High blood pressure | Systolic blood pressure ≥130 and/or diastolic blood pressure ≥85 mm Hg |
| | OR use of antihypertensive medication |
| Low high-density lipoprotein cholesterol | <40 mg/dL (1.0 mmol/L) in males |
| | <50 mg/dL (1.3 mmol/L) in females |
| | OR use of cholesterol medication |
| High fasting plasma glucose | ≥100 mg/dL |
| | OR use of insulin or other antidiabetic medication |
| High fasting triglycerides[†] | ≥150 mg/dL |

[†]Analyses of elevated fasting triglycerides restricted to participants that did not report current cholesterol medication use

Anthropometrics and blood samples were taken in the Mobile Examination Center (MEC) according to standardized protocol. NHANES has survey weights that apply to the subsample of participants who participated in the MEC exams. The NHANES anthropometric survey collected data on waist circumference (in centimeters, cm) and blood pressure (in mm Hg) [25]. Blood pressure was measured three consecutive times after a five-minute rest. We used the average of the second and third readings [26] to calculate systolic and diastolic blood pressure. High density lipoprotein (HDL-C, mg/dL) was measured in venous blood.

Additionally, in the laboratory subsample fasting blood-based biomarkers were collected from participants who reported in the morning session after an overnight fast; additional survey weights account for the fasted laboratory subsample. Fasting plasma glucose (FPG) and fasting triglycerides were measured in this blood panel and were available in mg/dL [26].

In addition to the continuous values, all variables were dichotomized using the criteria of cardiometabolic risk in the definition of Metabolic Syndrome (MetS) [24] (Table 2).

## Covariates

All sociodemographic information was self-reported as part of a standardized questionnaire. Age data were modeled in ten-year age categories. Income data were classified using the Poverty Income Ratio (PIR), a measure of family income relative to the Federal Poverty Level that accounts for household size. Income was categorized as PIR 0–185%, PIR 186–399%, PIR ≥ 400%, and Missing (due to high missingness in self-reported income, 8.1%) [27]. Education was reported in continuous years and classified as high school equivalent or lower, some college, and college degree or higher [28]. Race/ethnicity data were self-reported via categorical selection and classified as Non-Hispanic white, Non-Hispanic Black, Hispanic, Non-Hispanic Asian, or Other race/ethnicity (including Multiracial) [27, 29].

## Statistical analyses

Because the three indices have different value ranges, in descriptive analyses, we rescaled each index to have a range of 0 to 100. Bland-Altman plots were used to evaluate systematic differences in the continuous index values [30]. Pearson's correlation coefficient was used to assess correlation of continuous values, and radar plots were used to visually inspect how individual

components contributed to overall index values. To examine differences in index score by sociodemographic characteristics, we used survey-weighted regression with the standardized index scores as the dependent variable and dummy variables for each level of a given sociodemographic characteristic (sex, age, income, education, race/ethnicity) as the independent predictor variables.

In additional descriptive analyses, participants were classified into quintiles for each diet index (PHDI, HEI-2015, and DASH). Survey-weighted tables were used to examine percent agreement between quintiles of the three dietary indices and to examine the distribution of sociodemographic characteristics across quintile of each dietary index.

To directly compare the dietary indices and to test for linear trends, we created a standardized Z-score variable for each index (mean of zero, standard deviation of 1) and included this variable as a continuous exposure in survey-weighted linear regression models. We also used survey-weighted logistic regression models to estimate the association between diet Z-score and each cardiometabolic risk factor dichotomized according to the Metabolic Syndrome criteria (high waist circumference, high blood pressure, low HDL-C, high fasting plasma glucose, high triglycerides). For both linear and logistic regressions, models were adjusted for age, sex, income, education, race/ethnicity, and total energy intake.

In addition to our main analyses, we conducted several sensitivity analyses. We repeated the main analyses using quintile of dietary pattern as the exposure. Stata's postestimation margins, dydx command was used to estimate the change in probability of outcome by quintile of dietary index. In additional sensitivity analyses, we systematically tested adding smoking behavior, alcohol use, and physical activity into our final model (S1 Methods). No combination of these additional covariates had a significant effect on model estimates, so they were excluded from the final models.

To mitigate concerns about reverse causality in participants who made dietary changes or began medication use after receiving advice from a physician, we conducted additional sensitivity analyses for all blood pressure, HDL-C, and FPG models restricted to participants who were not currently taking medication and who had never been diagnosed with the respective risk factor (i.e., high blood pressure, low HDL-C, and high FPG) by a doctor.

All analyses were conducted in Stata 17.0 and $p < 0.05$ was considered significant.

## Results and discussion

### Results

The final sample size was 8,128 participants for the laboratory-based sample and 3,933 participants for the fasted subsample (Table 3). The survey-weighted prevalence of cardiometabolic risk factors ranged from 36.6% (95% CI: 34.1, 39.1%) for low HDL-C to 62.4% (59.8, 65.0%) for high FPG. The range of PHDI values was 21–125 on a scale from 0 to 140, whereas HEI-2015 values ranged from 15 to 99 on a scale of 0–100, and DASH spanned the theoretical range of 8 to 40. All three dietary indices were approximately normally distributed.

For continuous index values, the unweighted correlation between HEI-2015 and DASH ($\rho$ = 0.78) was slightly stronger than that of PHDI and DASH ($\rho$ = 0.66) or PHDI and HEI-2015 ($\rho$ = 0.65). The Bland-Altman plots of differences for each pairwise comparison of values are shown in Fig 1. In survey-weighted tables, 45.8% (41.4, 50.4%) of those in the lowest quintile of HEI-2015 were in the lowest quintile of PHDI, 50.7% (44.1, 57.3%) in the lowest quintile of DASH and PHDI, and 62.8% (57.4, 67.9%) of those in the lowest quintile of HEI-2015 were also in the lowest quintile of DASH (Fig 2). For the highest quintile, the concordance was 61.6% (57.2, 65.9%) for PHDI and DASH, 54.4% (49.1, 59.5%) for PHDI and DASH, and 69.0% (62.0, 75.1%) for HEI-2015 and DASH. When looking at all three indices, concordance

**Table 3. Characteristics of eligible participants with two days of dietary recall data, NHANES 2015–2018\*.**

| | |
|---|---|
| **Sex** | |
| Male | 49.1 (3954) |
| Female | 50.9 (4174) |
| **Mean age (SD)**, years | 48.6 (15.6) |
| **Educational attainment** | |
| High school equivalent or lower | 35.5 (3425) |
| Some college | 32.1 (2575) |
| College degree or greater | 32.4 (2121) |
| **Income** | |
| Poverty-to-Income Ratio < 185% | 28.6 (3212) |
| Poverty-to-Income Ratio 185–399% | 28.3 (2217) |
| Poverty-to-Income Ratio ≥ 400% | 35.0 (1874) |
| Missing income information | 8.1 (825) |
| **Race/ethnicity** | |
| Non-Hispanic white | 64.1 (2949) |
| Non-Hispanic Black | 11.1 (1873) |
| Hispanic | 14.8 (2054) |
| Asian, Multiracial, and Other Non-Hispanic race/ethnicities | 10.0 (1252) |
| **Mean (SD) PHDI** | 62 (54–70) |
| **Mean (SD) HEI-2015** | 53 (44–63) |
| **Mean (SD) DASH** | 24 (19–28) |
| **Prevalence of cardiometabolic risk factors** | |
| Elevated waist circumference | 61.0 (4815) |
| Elevated blood pressure | 43.8 (4132) |
| Reduced high density lipoprotein cholesterol (HDL-C) | 41.7 (3535) |
| Elevated fasting triglycerides[†] | 36.6 (1672) |
| Elevated fasting glucose[†] | 62.4 (2460) |

\* Values are weighted % (unweighted N) unless otherwise noted. Weighted % accounts for complex survey weights.

[†] Results are from fasted subsample only and reflect use of fasted analytic weights.

was 34.7% (30.5, 39.2%) for the lowest quintile–meaning that of participants in quintile 1, 34.7% of participants were in the quintile 1 for all three dietary values–and 41.4% (36.6, 46.4%) for the highest quintile.

We observed several disparities in diet quality (Table 4). For all three rescaled dietary indices, mean dietary quality was lower for men than for women, and tended to be lower for younger individuals. People with low income and low education, as well as individuals who identified as Non-Hispanic Black, also had lower dietary quality as measured by all three indices. For PHDI and DASH only, there was also a significant gradient in dietary quality across income category. Finally, individuals who identified as Hispanic had lower dietary scores as measured by PHDI or DASH, but not for HEI-2015.

A higher score on all three dietary indices was associated with health-promoting differences in cardiometabolic risk factors. Waist circumference decreased by a range of 1.5 (0.5, 2.5) centimeters per 1-SD increase in PHDI to 2.5 (1.8, 3.2) centimeters per 1-SD increase in DASH (Table 5). We observed comparable results using the binary risk factor thresholds: risk of high waist circumference decreased by 3.8 (1.9, 5.7), 4.4 (2.2, 6.5) and 4.7 (2.5, 7.0) percentage points per 1-SD increase in the PHDI, HEI-2015, and DASH values, respectively.

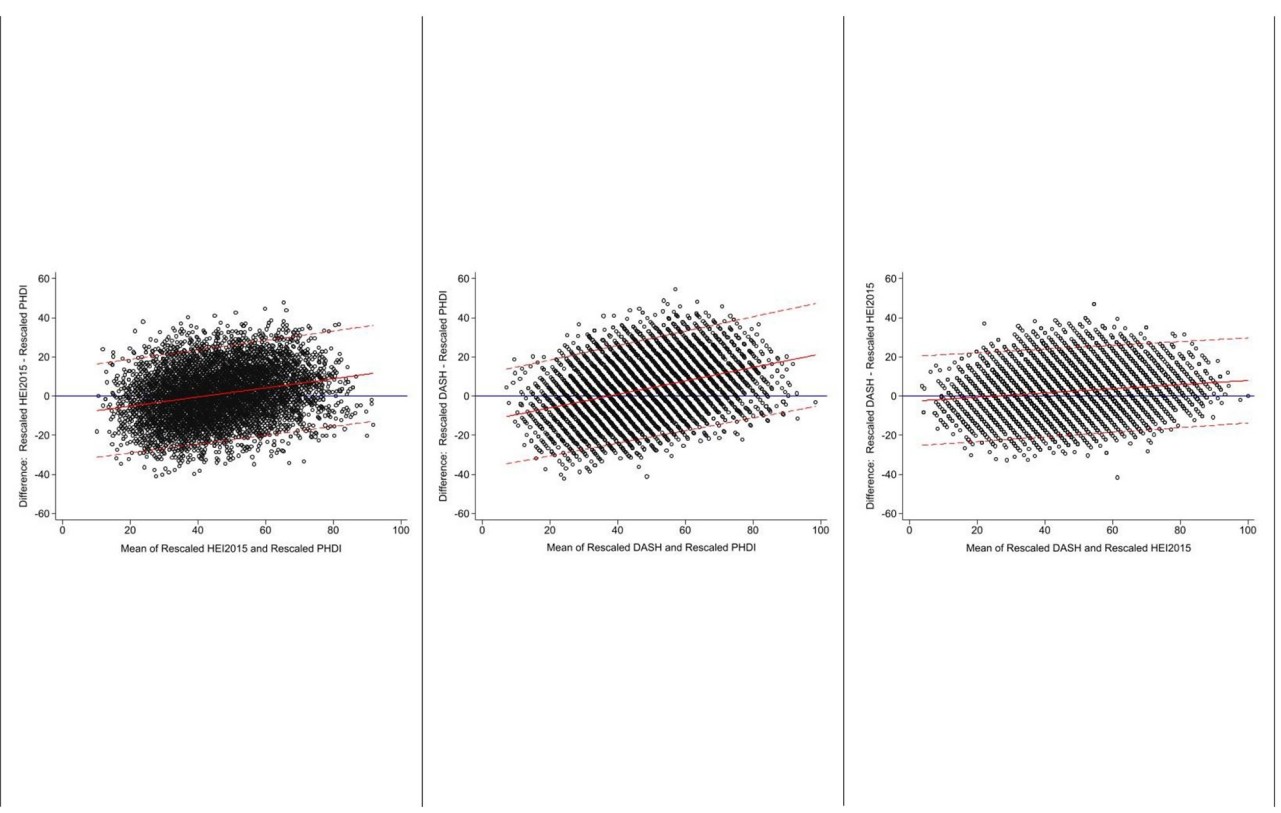

**Fig 1. Bland-Altman plots comparing rescaled PHDI, HEI-2015, and DASH values.** Planetary Health Diet Index, Healthy Eating Index-2015, and Dietary Approaches to Stop Hypertension scores were rescaled from 0 to 100 for comparability.

For blood pressure, a 1-SD increase in PHDI and HEI-2015 scores were associated with lower systolic blood pressure, but not with lower diastolic blood pressure (Table 5). Higher DASH z-score was associated with lower systolic and diastolic blood pressure. In logistic regression, the predicted probability of high blood pressure decreased across the three indices, ranging from a reduction of 2.9 (0.6, 5.2) percentage points for a 1-SD increase in PHDI to 3.9 (2.2, 5.6) percentage points for DASH.

All three dietary indices were associated with higher HDL-C, ranging from 1.5 (0.9, 2.1) mg/dL higher for a 1-SD increase in DASH to 2.1 (1.6, 2.5) mg/dL higher for HEI-2015 (Table 5). The predicted probability of low HDL-C decreased by a range of 2.9 (1.0, 4.8) percentage points for a 1-SD increase in DASH to 4.3 (2.5, 5.8) percentage points for every 1-SD increase in HEI-2015.

In the fasted subsample, there were no significant associations between dietary index z-score and FPG (Table 5). For the logistic regression analyses using the MetS cutoffs, the predicted probability of high FPG decreased by 2.8 (0.1, 4.8) percentage points for a 1-SD increase in HEI-2015 and 2.4 (0.3, 4.5) percentage points per 1-SD increase in DASH. We did not observe a significant association between PHDI and the binary high FPG outcome.

For fasting triglycerides, a 1-SD increase in DASH was associated with lower fasting triglycerides (Table 5). PHDI and HEI-2015 were not associated with continuous fasting triglycerides. We did not observe a significant association between any of the dietary indices and predicted probability of elevated fasting triglycerides.

### HEI-2015

|  | | Quintile 1 | Quintile 2 | Quintile 3 | Quintile 4 | Quintile 5 |
|---|---|---|---|---|---|---|
| **PHDI** | Quintile 1 | **9.97** | 5.88 | 3.47 | 1.25 | 0.18 |
| | Quintile 2 | 5.98 | **6.35** | 4.79 | 2.88 | 0.98 |
| | Quintile 3 | 2.97 | 4.81 | **5.77** | 4.24 | 2.10 |
| | Quintile 4 | 1.22 | 2.78 | 4.56 | **5.52** | 4.61 |
| | Quintile 5 | 0.17 | 0.97 | 2.67 | 5.16 | **10.70** |

### DASH

|  | | Quintile 1 | Quintile 2 | Quintile 3 | Quintile 4 | Quintile 5 |
|---|---|---|---|---|---|---|
| **PHDI** | Quintile 1 | **10.67** | 6.21 | 2.56 | 1.07 | 0.25 |
| | Quintile 2 | 6.31 | **6.75** | 4.12 | 2.55 | 1.24 |
| | Quintile 3 | 2.82 | 5.81 | **5.04** | 3.67 | 2.56 |
| | Quintile 4 | 1.06 | 3.74 | 4.33 | **4.88** | 4.69 |
| | Quintile 5 | 0.23 | 1.22 | 2.79 | 4.17 | **11.26** |

### HEI-2015

|  | | Quintile 1 | Quintile 2 | Quintile 3 | Quintile 4 | Quintile 5 |
|---|---|---|---|---|---|---|
| **DASH** | Quintile 1 | **12.71** | 5.92 | 2.07 | 0.39 | 0.00 |
| | Quintile 2 | 5.30 | **7.96** | 7.17 | 2.93 | 0.37 |
| | Quintile 3 | 1.83 | 4.28 | **6.14** | 5.09 | 1.50 |
| | Quintile 4 | 0.39 | 2.05 | 4.12 | **5.68** | 4.08 |
| | Quintile 5 | 0.06 | 0.58 | 1.76 | 4.96 | **12.64** |

**Cross-tabulation**

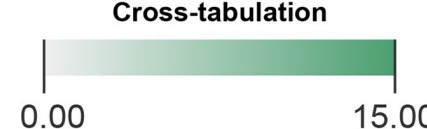

0.00 15.00

**Fig 2. Percent agreement for quintiles of PHDI, HEI-2015 and DASH, NHANES 2015–2018.** Values are percent in a given quintile of one index that are in the same quintile of the other index. Perfect correlation would be 20.00% down the diagonal.

**Table 4. Predicted standardized PHDI, HEI-2015, and DASH value by sociodemographic characteristics, NHANES 2015–2018[*],[†].**

| | PHDI | HEI-2015 | DASH |
|---|---|---|---|
| *Sex* | | | |
| Male[‡] | 44.0 (42.8, 45.1) | 44.2 (42.8, 45.5) | 46.0 (44.4, 47.5) |
| Female | 47.1*** (46.1, 48.2) | 47.2*** (45.6, 48.7) | 51.7*** (49.8, 53.5) |
| *Age category* | | | |
| 20–29[‡] | 43.0 (41.5, 44.5) | 41.4 (39.4, 43.4) | 43.5 (41.4, 45.6) |
| 30–39 | 45.0 (43.3, 46.6) | 43.9** (42.1, 45.7) | 45.9* (43.8, 48.1) |
| 40–49 | 44.9* (43.8, 46.0) | 45.5*** (44.4, 46.7) | 48.1*** (46.3, 49.8) |
| 50–59 | 46.1** (44.2, 48.1) | 46.8*** (44.6, 49.1) | 49.6*** (46.9, 52.3) |
| 60–69 | 47.1*** (45.7, 48.5) | 48.2*** (46.3, 50.0) | 52.2*** (50.4, 54.0) |
| 70–79 | 48.3*** (46.8, 49.8) | 49.7*** (48.0, 51.3) | 55.3*** (53.5, 57.1) |
| 80 or older | 47.1*** (45.6, 48.6) | 48.5*** (46.3, 50.6) | 56.1*** (54.0, 58.3) |
| *Income* | | | |
| PIR < 185% [‡] | 42.5 (41.4, 43.6) | 42.3 (41.0, 43.6) | 44.5 (42.9, 46.1) |
| PIR 185–399% | 45.0*** (43.7, 46.3) | 44.9*** (43.3, 46.6) | 48.1*** (46.2, 50.0) |
| PIR ≥ 400% | 48.5*** (47.3, 49.7) | 48.9*** (47.1, 50.7) | 53.0*** (50.9, 55.1) |
| Missing | 45.7** (43.7, 47.6) | 46.4*** (44.0, 48.8) | 49.2*** (46.6, 51.2) |
| *Education* | | | |
| High school or lower[‡] | 42.4 (41.5, 43.4) | 42.0 (40.6, 43.4) | 43.8 (42.1, 45.4) |
| Some college | 44.1** (42.9, 45.3) | 43.9* (42.3, 45.5) | 46.9*** (45.0, 48.8) |
| College degree or greater | 50.5*** (49.3, 51.7) | 51.5*** (50.0, 53.1) | 56.5*** (54.8, 58.1) |
| *Race/ethnicity* | | | |
| Non-Hispanic white[‡] | 46.2 (45.1, 47.3) | 45.7 (44.2, 47.2) | 49.9 (48.2, 51.6) |
| Non-Hispanic Black | 40.3*** (39.3, 41.4) | 42.5*** (40.9, 44.2) | 41.8*** (40.0, 43.6) |
| Hispanic | 44.4** (43.3, 45.4) | 45.3 (43.8, 46.9) | 47.9* (46.3, 49.5) |
| Asian, Multiracial, and Other Non-Hispanic | 49.0*** (47.3, 50.7) | 49.7*** (47.9, 51.6) | 51.6 (49.2, 53.9) |

[*] Distribution of dietary scores were standardized to 0 to 100 scale for each index.

[†] Values are from linear regression with standardized continuous score (range: 0–100) as the dependent variable and dummy indicators for sociodemographic category as independent variables.

[‡] Indicates reference category

* p<0.05, **p<0.01, ***p<0.001 for the difference from the referent category

In sensitivity analyses of participants who had not been previously diagnosed with the given risk factor, the pattern of results was consistent with the main analyses for blood pressure (N = 4921) and HDL-C (N = 4580, S2 Table). For continuous results of FPG (N = 3094), there was still a negative association between higher dietary index score and lower FPG for all three indices, although the magnitude of the results was attenuated. Additionally, in the sensitivity analyses for FPG, higher PHDI was associated with a lower predicted probability of high FPG (S2 Table). Logistic regression using quintiles of PHDI as the exposure did not substantively impact our conclusions (S2 Fig, S3 Table).

## Discussion

To our knowledge, this is the first study to compare a dietary index created with both health and environmental considerations, the PHDI, to two frequently used dietary indices created

**Table 5. Predicted change in continuous and binary cardiometabolic risk factors per one standard-deviation change in PHDI, HEI-2015, and DASH, NHANES 2003–2018\*.**

| | PHDI | HEI-2015 | DASH | p-value[‡] |
|---|---|---|---|---|
| *Waist circumference* | | | | |
| Centimeters | -1.9\*\*\* (-2.5, -1.2) | -2.3\*\*\* (-3.0, -1.5) | -2.5\*\*\* (-3.2, -1.8) | 0.03 |
| Predicted probability of high waist circumference | -3.8\*\*\* (-5.7, -1.9) | -4.4\*\*\* (-6.5, -2.2) | -4.7\*\*\* (-7.0, -2.5) | 0.54 |
| *Blood pressure* | | | | |
| Systolic blood pressure, mm Hg | -0.5 (-1.2, -0.1) | -0.9\*\* (-1.5, -0.4) | -1.2\*\*\* (-1.7, -0.6) | 0.34 |
| Diastolic blood pressure, mm Hg | -0.2 (-0.7, 0.2) | -0.5 (-1.1, 0.1) | -0.7\* (-1.3, -0.2) | 0.49 |
| Predicted probability of high blood pressure | -2.9\* (-5.2, -0.6) | -3.7\*\* (-5.7, -1.7) | 3.9\*\* (-5.6, -2.1) | 0.60 |
| *High-density lipoprotein cholesterol, HDL-C* | | | | |
| mg/dL | 1.9\*\*\* (1.3, 2.5) | 2.1\*\*\* (1.6, 2.5) | 1.5\*\*\* (0.9, 2.1) | 0.20 |
| Predicted probability of low HDL-C | -4.2\*\*\* (-5.8, 2.6) | -4.3\*\*\* (-5.8, -2.8) | -2.9\*\* (-4.8, -1.0) | 0.19 |
| *Fasting plasma glucose, FPG* | | | | |
| mg/dL | -0.2 (-1.2, 0.8) | -0.3 (-1.7, 1.1) | 0.0 (-1.6, 1.6) | 0.64 |
| Predicted probability of high FPG | -2.3 (-4.8, 0.0) | -2.8\*\* (-4.8, -0.1) | -2.4\* (-4.5, -0.3) | 0.71 |
| *Fasting triglycerides* | | | | |
| mg/dL[†] | -4.6\* (-9.2, -0.1) | -3.7\* (-8.0, -0.5) | -5.4\* (-9.3, -1.4) | 0.59 |
| Predicted probability of high fasting triglycerides | -1.8 (-4.1, 0.0) | 0.9 (-3.6, 1.8) | -1.0 (-3.4, 1.4,) | 0.66 |

\* Survey-weighted regression models were adjusted for age, sex, income, education, race/ethnicity, and total energy intake.

[†] \*p<0.05, \*\*p<0.01, \*\*\*p<0.001 for the difference from 0 as estimated by a Wald test.

[‡] P value for the joint comparison of the three indices as estimated by a Wald test.

with health considerations only (HEI-2015 and DASH). We found a moderate correlation between the indices, with HEI-2015 and DASH more strongly correlated with each other than with PHDI. As expected, across the indices, higher diet quality was correlated with lower predicted probability of cardiometabolic risk across the risk factors examined here. Importantly, our results from the US are consistent with analyses of EAT-*Lancet* style dietary patterns in other countries that have found that a higher intake of this dietary pattern was associated with lower risk of type II diabetes in Mexico [31], the UK [5], and Denmark [32] and lower prevalence of cardiometabolic risk in the UK [5] and Brazil [33]. Finally, we find that disparities in diet quality are consistent across the three indices, highlighting the need for policies to promote access to healthy diets for vulnerable populations in the US.

This study is among the first to examine how a dietary pattern that measures adherence to the EAT-*Lancet* guidelines, the PHDI, compares to two well-established ways of measuring healthy diets. All three dietary indices share some common elements, such as encouraging high intakes of fruit, vegetables, and whole grains, and discouraging intake of added sugar and saturated fat. Yet of the three indices examined here, population-level distribution of PHDI values was lowest, and on the Bland-Altman plots were consistently lower than either HEI-2015 or DASH. This is likely because HEI-2015 is designed to reflect adherence to the Dietary Guidelines for Americans that were created to promote health within the American cultural context, and because DASH is designed to reflect hypertension risk, but its values are derived based on the distribution of intake in the underlying NHANES population. In contrast, PHDI is intended as a global reference diet that incorporates both diet and environmental risk using pre-defined cutpoints.

With this context in mind, the different ways that HEI-2015, DASH, and PHDI treat food groups makes the same diet score differently. For example, PHDI discourages starchy

vegetables, emphasizes a high intake of plant sources of proteins such as legumes, nuts and seeds and has stricter scoring criteria for added sugars and saturated/trans fats than do HEI-2015 or DASH, such that the median value for these components was zero on the PHDI. Both HEI-2015 and DASH consider starchy vegetables under the encouraged total vegetable component. HEI-2015 scoring does not use mutually-exclusive categories and triple counts beans and legumes in the total vegetables, greens and beans, and seafood and plant proteins components [8], leading to higher HEI-2015 values for the same quantity of food. Additionally, PHDI recommends a maximum of 14 grams of red and processed meat intake per day. But the median value on the PHDI red and processed meat component was 5 out of 10, and the median intake of red and processed meat was over four times that of the PHDI recommendations, at 62 grams. HEI-2015, on the other hand, does not place an upper limit on meat intake and in fact encourages it in the total protein foods component, whose median value was the maximum 5 out of 5 points. Taken together, the differences in index construction, in scoring criteria for added sugars and saturated/trans fats, and in the conceptualization of red and processed meat as a discouraged or an encouraged component could explain the differences in the distribution of PHDI, HEI-2015, and DASH scores observed in our descriptive analyses.

Despite these differences, PHDI, HEI-2015, and DASH performed comparably in our primary analyses. First, American dietary quality according to each index was well below the theoretical maximum, aligning with other studies which similarly find that the average diets of Americans do not conform to dietary recommendations. Second, and most importantly, higher dietary quality as measured by each of these indexes is associated with lower cardiometabolic risk factors [10, 34]. Third, the indices performed comparably with respect to correlations with the cardiometabolic risk factors we examined, although PHDI was the only index that was associated with lower risk of elevated fasting triglycerides and was not as strongly associated with blood pressure when comparing intake quintiles. For triglycerides, this could be due to the inclusion of starchy vegetables as a separate, discouraged component in PHDI as well as a lower maximum saturated fat value. Both high intake of low-glycemic foods and saturated fats are associated with high triglycerides [35, 36]. On the other hand, PHDI does not have a sodium component where the other two indices do, and high sodium intake is a strong predictor of high blood pressure [37]. Despite these differences, all three diets have healthy plant-based options, which have not only been associated with lower cardiometabolic risk in a large US-based cohort study, but also have significant benefits for environmental sustainability [38].

We also observed disparities in diet quality across the three indices, such that populations that were Black or that had low levels of income or education had poorer diet quality. The disparities for PHDI were consistent with those observed for HEI-2015 and for DASH. Indeed, diet disparities in the US have been well-documented [12, 39, 40] and are tied to a combination of physical, social, economic, and political factors that make it difficult to access and afford healthy food [41]. Due to these structural factors, vulnerable populations in the US will also be disproportionately impacted by increases in food prices caused by climate change, exacerbating disparities in both food security and dietary quality [42]. These populations are also more susceptible to other threats to health and livelihood caused by climate change, again due to systematic inequalities that increase their risk of exposure to climate events and negatively impact their capacity to adapt [43, 44]. Ideally, policy solutions would address upstream determinants of health disparities and would lead to improvements in dietary quality measured by PHDI, HEI-2015, and DASH. But from a holistic health perspective, addressing disparities in PHDI–which is designed to address both nutritional and environmental aspects of long-term health–could have even greater benefits than using an index that considers nutrition alone.

### Limitations and strengths

The present study has several limitations. Twenty-four hour recall data are subject to measurement error and do not represent usual intake. However, we use data from two days of dietary recall to obtain more information on participants' diets and restricted our sample to participants with plausible total energy intakes. Additionally, PHDI is scored based on fixed intakes for a 2500 kilocalorie/day diet, while HEI-2015 and DASH use the energy density approach for scoring. NHANES is a cross-sectional survey, so we cannot establish causal inference for long-term disease risk, and reverse causality is possible. We did, however, conduct rigorous sensitivity analyses in undiagnosed participants, which mitigate concerns about dietary changes made at the advice of a physician.

This study also has several strengths. It is the first to use nationally-representative data to examine the correlation between the EAT-*Lancet* Commission's dietary recommendations and cardiometabolic risk factors in the US. It also provides valuable context by directly comparing the PHDI with two other well-established dietary indices.

### Conclusion

Our analysis suggests that sustainability-focused dietary recommendations, which we operationalized using the PHDI, have similar benefits for cardiometabolic risk factors as HEI-2015 and DASH. There is a need for effective policy solutions to support healthy diets overall, and particularly for populations suffering from a high burden of diet-related disease. Including sustainability in dietary guidelines can have environmental co-benefits while promoting population-level cardiometabolic health.

### Supporting information

**S1 Methods.**
(DOCX)

**S1 Table. Scoring criteria for the Planetary Health Diet Index (PHDI).** [*] Grams per day calculated from dry weight. [†] To calculate the score for the legumes component, the non-soy and soy subcomponents are each weighted at 0.5.
(DOCX)

**S2 Table. Predicted change in continuous and binary cardiometabolic risk factors per one standard-deviation score in Planetary Health Diet Index, Healthy Eating Index-2015, and Dietary Approaches to Stop Hypertension score among undiagnosed participants only, National Health and Nutrition Examination Survey 2003–2018[\*].** [*] Survey-weighted regression models were adjusted for age, sex, income, education, race/ethnicity, and total energy intake. [†] mg/dL = milligrams per deciliter.
(DOCX)

**S3 Table. Predicted probability of cardiometabolic risk factor by quintile of Planetary Health Diet Index, Healthy Eating Index-2015, and Dietary Approaches to Stop Hypertension score, National Health and Nutrition Examination Survey 2003–2018[\*,†].** [*] Survey-weighted logistic regression models were adjusted for age, sex, income, education, race/ethnicity, and total energy intake. [†] * $p<0.05$, ** $p<0.01$, *** $p<0.001$. [‡] Contrast is from Stata's post-estimation margins, dydx command and represents percentage point reduction in predicted probability from Quintile 1 to Quintile 5.
(DOCX)

**S4 Table. Predicted probability of cardiometabolic risk factor by quintile of Planetary Health Diet Index, Healthy Eating Index-2015, and Dietary Approaches to Stop Hypertension value among undiagnosed participants only, National Health and Nutrition Examination Survey 2003–2018\*,†.** \* Survey-weighted logistic regression models were adjusted for age, sex, income, education, race/ethnicity, and total energy intake.† \* p<0.05, \*\* p<0.01, \*\*\* p<0.001. ‡ Contrast is from Stata's postestimation margins, dydx command and represents percentage point reduction in predicted probability from Quintile 1 to Quintile 5.
(DOCX)

**S1 Fig. Radar plots of median component scores for Planetary Health Diet Index (PHDI), Healthy Eating Index-2015 (HEI-2015), and Dietary Approaches to Stop Hypertension (DASH), National Health and Nutrition Examination Survey 2015–2018.** \* All dietary pattern component scores range 0–10 unless otherwise noted. † Component score range: 0–5.
(TIF)

**S2 Fig. Estimated change in predicted probability of cardiometabolic risk factors between Quintiles 1 and 5 of Planetary Health Diet Index, Healthy Eating Index-2015, and Dietary Approaches to Stop Hypertension score\*,†.** \* Logistic regression models were adjusted for age, sex, income, education, and race/ethnicity. \* p<0.05, \*\* p<0.01, \*\*\* p<0.001 for the estimated contrast between Quintile 1 and Quintile 5.
(TIF)

**S1 Checklist. STROBE statement—Checklist of items that should be included in reports of observational studies.**
(DOCX)

## Author Contributions

**Conceptualization:** Sarah M. Frank.

**Data curation:** Sarah M. Frank.

**Formal analysis:** Sarah M. Frank.

**Funding acquisition:** Lindsay M. Jaacks, Lindsey Smith Taillie.

**Supervision:** Lindsay M. Jaacks, Lindsey Smith Taillie.

**Visualization:** Sarah M. Frank.

**Writing – original draft:** Sarah M. Frank.

**Writing – review & editing:** Sarah M. Frank, Lindsay M. Jaacks, Christy L. Avery, Linda S. Adair, Katie Meyer, Donald Rose, Lindsey Smith Taillie.

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
