## [Decision Letter · Decision Letter 0]

11 Jul 2023

PONE-D-23-16525Dietary quality and cardiometabolic indicators in the USA: A comparison of the Planetary Health Diet Index, Healthy Eating Index-2015, and Dietary Approaches to Stop HypertensionPLOS ONE

Dear Dr. Frank,

Thank you for submitting your manuscript to PLOS ONE. After careful consideration, we feel that it has merit but does not fully meet PLOS ONE’s publication criteria as it currently stands. Therefore, we invite you to submit a revised version of the manuscript that addresses the points raised during the review process. Kindly exert all the suggestions of the reviewers and academic editor of the manuscript so as not to be in the process of re-review or rejection of the manuscript. All applied changes should be yellow highlighted.

A rebuttal letter that responds to each point raised by the academic editor and reviewer(s). You should upload this letter as a separate file labeled 'Response to Reviewers'.A marked-up copy of your manuscript that highlights changes in YELLOW made to the original version. You should upload this as a separate file labeled 'Revised Manuscript with Track Changes'.An unmarked version of your revised paper without tracked changes. You should upload this as a separate file labeled 'Manuscript'.

We look forward to receiving your revised manuscript.

Kind regards,

Mohammad Reza Mahmoodi, Ph.D.

Academic Editor

PLOS ONE

Journal Requirements:

3. Please ensure that you include a title page within your main document. 

Reviewers' comments:

Reviewer's Responses to Questions

**Comments to the Author**

1. Is the manuscript technically sound, and do the data support the conclusions?

Reviewer #1: Partly

Reviewer #2: Yes

2. Has the statistical analysis been performed appropriately and rigorously? 

Reviewer #1: Yes

Reviewer #2: Yes

3. Have the authors made all data underlying the findings in their manuscript fully available?

Reviewer #1: Yes

Reviewer #2: Yes

4. Is the manuscript presented in an intelligible fashion and written in standard English?

Reviewer #1: Yes

Reviewer #2: Yes

5. Review Comments to the Author

Reviewer #1: Comments to authors:

The main objective of this study was to explore the association between PHDI, HEI, and DASH with cardiometabolic biomarkers. The authors revealed that the association between HEI and DASH with the majority of cardiometabolic biomarkers was stronger than the association between PHDI with cardiometabolic biomarkers.

1. Another very interesting topic that has been addressed by dear authors but not deeply focused on has been socio-economic disparities. Other researchers have shown that the more people deviate from a healthy eating pattern, the more cardiometabolic biomarkers worsen both in people with risk factors such as diabetes and in healthy people. This has been well-established in many studies on food insecurity. In many communities where people are food insecure, their cardiometabolic biomarkers have worsened. In very simple language, food insecurity means insufficiency and imbalance in receiving all food groups in the daily food pattern.

2. Authors should know another point well that all parts of a manuscript/article must follow a certain homogeneity and respond to the main objectives of the study. Therefore, the authors are strongly requested to do the same in the discussion of the manuscript as they showed the correlation difference between the three indicators with selected biomarkers in the results of the study.

3. Another point that the authors know very well is that more than 26 years have been studied on the healthy eating index and more than 20 years on the DASH model, and these two patterns have well proven their role in the nutritional health of the people of a society. Therefore, avoid one-sidedness in the discussion which is the most essential part of the article.

4. The researchers who initially designed, implemented, and presented the healthy eating index and DASH aimed to implement the policies to promote access to healthy diets for vulnerable populations in the US.

5. The strengths and limitations section should be presented in a separate section under the subtitle of “the strengths and limitations” before the conclusion.

6. The last point is that the conclusion at the end of the article should be modified according to the obtained results.

Reviewer #2: I would thank the authors for this valuable article. I have some questions:

1- In Table 1, please say why you report the three indices (PHDI, HEI, and DASH) with median (IQR) instead of Mean (SD).

2- In Table 5, You did not report any p-values for linear and logistic regressions. Please add p-values to test whether, for example, high waist circumference risk significantly differs in the three indices.

6. PLOS authors have the option to publish the peer review history of their article (what does this mean?). If published, this will include your full peer review and any attached files.

Reviewer #1: No

Reviewer #2: No

---

## [Author Response · Author response to Decision Letter 0]

12 Sep 2023

Journal Requirements:

https://journals.plos.org/plosone/s/file?id=ba62/PLOSOne_formatting_sample_title_authors_affiaffiliat.pdf

Thank you. We have followed the example formatting above for the manuscript and for the title page. 

Thank you. We have updated the language as follows: “The study protocols of the NHANES are approved by the Research Ethics Review Board at the National Center for Health Statistics (NCHS) [14]. This is a retrospective study of data that were fully-anonymized before the authors accessed them. Because the de-identified observational data from the National Health and Nutrition Examination Survey are publicly available for download, this study received a determination of Not Human Subjects Research by the Institutional Review Board at [First Author’s Home University]."

3. Please ensure that you include a title page within your main document. 

Thank you. We have included a title page in the main document and formatted according to the guidelines available at the link provided by the Journal.

Thank you for this feedback. The cited results are not a core part of the research being presented in the study and the phrase “data not shown” has been removed. 

Thank you. We have included the captions for our Supporting Information files at the end of the manuscript. We have additionally uploaded each Supporting Information file individually per the guidelines on the PLOS ONE website.

Reviewer #1: Comments to authors:

The main objective of this study was to explore the association between PHDI, HEI, and DASH with cardiometabolic biomarkers. The authors revealed that the association between HEI and DASH with the majority of cardiometabolic biomarkers was stronger than the association between PHDI with cardiometabolic biomarkers.

1. Another very interesting topic that has been addressed by dear authors but not deeply focused on has been socio-economic disparities. Other researchers have shown that the more people deviate from a healthy eating pattern, the more cardiometabolic biomarkers worsen both in people with risk factors such as diabetes and in healthy people. This has been well-established in many studies on food insecurity. In many communities where people are food insecure, their cardiometabolic biomarkers have worsened. In very simple language, food insecurity means insufficiency and imbalance in receiving all food groups in the daily food pattern.

Thank you for this comment. We wholeheartedly agree that food insecurity contributes to dietary disparities and to poorer cardiometabolic outcomes, particularly for historically disadvantaged populations. An analysis of food insecurity was beyond the scope of the present manuscript. We did find disparities in dietary quality for all three indices (PHDI, HEI-2015, and DASH) by key sociodemographic indicators, including income, education, and race/ethnicity, and reported on these in the results section. In the discussion we talk about the disparities in dietary quality that were observed in our study and that are consistent with dietary disparities that have been documented repeatedly in the literature (lines 331-332). We also mention that these disparities are due to a variety of structural factors that make accessibility and affordability of health food difficult, and that climate change has the potential to exacerbate these disparities (lines 332-339). We then mention that policy solutions are needed to address upstream determinants of health disparities and improve dietary quality (lines 339-341). 

In this way, we hope that our study contributes to the important conversation about disparities in dietary quality, particularly for historically disadvantaged populations, and adds to the calls for policy solutions to ensure equitable access to healthy diets for all Americans. 

2. Authors should know another point well that all parts of a manuscript/article must follow a certain homogeneity and respond to the main objectives of the study. Therefore, the authors are strongly requested to do the same in the discussion of the manuscript as they showed the correlation difference between the three indicators with selected biomarkers in the results of the study.

Thank you for this comment. We believe that the discussion does follow a standard manuscript format, as we first summarize our results, then qualitatively compare the indices (which aligns with our descriptive analyses), summarize and discuss the implications of the similar correlations with cardiometabolic (which aligns with our primary objective and main regression analyses), discuss SD (which aligns with our secondary objective), and then move on to strengths and limitations. We have added some language to the discussion to signal which part of the results we are referring to in a given paragraph (e.g. “our descriptive analyses”, “our primary analyses”). 

Please note that because of the similarities between the indices for our primary results, and because of the sheer quantity of results, we did not use discussion space to go through each biomarker individually. We believe such an approach would have been redundant for a reader and distracting from the other points we raised in the discussion. 

3. Another point that the authors know very well is that more than 26 years have been studied on the healthy eating index and more than 20 years on the DASH model, and these two patterns have well proven their role in the nutritional health of the people of a society. Therefore, avoid one-sidedness in the discussion which is the most essential part of the article.

Thank you for your comment. We agree with the reviewer that HEI and especially DASH are valuable public health nutrition tools. We spend more time discussing the PHDI since it is a novel index and may not be familiar to readers as HEI or DASH, whose benefits are already well-established. 

In the revision, we present a more balanced discussion of the three indices in several ways: 

• First, in the second and third paragraphs that compare PHDI, HEI-2015, and DASH scoring, we have emphasized that we are comparing the PHDI to two well-established ways of measuring healthy diets (lines 284-285). We have also reworded to emphasize “differences” between the dietary indices (lines 295-296) rather than making comparisons of “worse quality” as in the previous version. 

• Second, we have taken out the sentence “Despite these differences, overall healthy plant-based diets – such as the PHDI - have been associated with lower cardiometabolic risk in a large US-based cohort study [40], suggesting that improved long-term adherence to the PHDI would similarly be associated with decreased cardiometabolic risk over time.” We replaced this sentence with “Despite these differences, all three diets have healthy plant-based options, which have not only been associated with lower cardiometabolic risk in a large US-based cohort study, but also have significant benefits for environmental sustainability [40].” (lines 326-328). 

• Finally, in the last paragraph of the discussion, we mention that ideal policy solutions “would address upstream determinants of health disparities and would lead to improvements in dietary quality measured by PHDI, HEI-2015, and DASH (lines 339-341).”

4. The researchers who initially designed, implemented, and presented the healthy eating index and DASH aimed to implement the policies to promote access to healthy diets for vulnerable populations in the US.

Thank you for this comment. We agree that both HEI and DASH could be leveraged to reduce dietary disparities, and acknowledge in both our Introduction section (lines 75-75) and Discussion section (lines 331-332) that there are well-documented disparities for both of these indices. We further mention that policies could lead to improvements in both HEI and DASH as well as in PHDI (lines 339-341). 

5. The strengths and limitations section should be presented in a separate section under the subtitle of “the strengths and limitations” before the conclusion.

We appreciate this suggestion. We have delimited “Limitations and strengths” as a subsection under the Results and discussion main heading. Please note that Results and discussion is now one heading rather than two per journal formatting guidelines (available from https://journals.plos.org/plosone/s/file?id=wjVg/PLOSOne_formatting_sample_main_body.pdf).

6. The last point is that the conclusion at the end of the article should be modified according to the obtained results.

Thank you. We agree with the reviewer’s comment. We have modified the conclusion to first focus on the primary results of the study (i.e., the similar correlations with cardiometabolic indicators) at a high-level, and then have a high-level summary of the need for policy solutions. The conclusion now reads: “Our analysis suggests that sustainability-focused dietary recommendations, which we operationalized using the PHDI, have similar benefits for cardiometabolic risk factors as HEI-2015 and DASH. There is a need for effective policy solutions to support healthy diets overall, and particularly for populations suffering from a high burden of diet-related disease. Including sustainability in dietary guidelines can have environmental co-benefits while promoting population-level cardiometabolic health.” (lines 359-364). 

Reviewer #2: I would thank the authors for this valuable article. I have some questions:

1- In Table 1, please say why you report the three indices (PHDI, HEI, and DASH) with median (IQR) instead of Mean (SD).

Thank you for this comment. We have updated the table to report the mean (SD) of the scores rather than the median (IQR).

2- In Table 5, You did not report any p-values for linear and logistic regressions. Please add p-values to test whether, for example, high waist circumference risk significantly differs in the three indices.

Thank you for this comment. We have added columns for the pairwise test of the beta coefficients from each regression model into table 5. We have also added asterisks indicating the difference from zero for each predicted difference provided in table 5, and updated the table legend accordingly.

---

## [Decision Letter · Decision Letter 1]

10 Oct 2023

PONE-D-23-16525R1Dietary quality and cardiometabolic indicators in the USA: A comparison of the Planetary Health Diet Index, Healthy Eating Index-2015, and Dietary Approaches to Stop HypertensionPLOS ONE

Dear Dr. Frank,

Thank you for submitting your manuscript to PLOS ONE. After careful consideration, we feel that it has merit but does not fully meet PLOS ONE’s publication criteria as it currently stands. Therefore, we invite you to submit a revised version of the manuscript that addresses the points raised during the review process.

Please apply the suggestions of the respected reviewer to present the p-values for the logistic regression statistical test in cardiometabolic biomarkers between the three dietary guidelines. The consent of the statistical expert reviewer of your manuscript will be the condition of acceptance of your article.

We look forward to receiving your revised manuscript.

Kind regards,

Mohammad Reza Mahmoodi, Ph.D.

Academic Editor

PLOS ONE

Journal Requirements:

Reviewers' comments:

Reviewer's Responses to Questions

**Comments to the Author**

1. If the authors have adequately addressed your comments raised in a previous round of review and you feel that this manuscript is now acceptable for publication, you may indicate that here to bypass the “Comments to the Author” section, enter your conflict of interest statement in the “Confidential to Editor” section, and submit your "Accept" recommendation.

Reviewer #1: All comments have been addressed

Reviewer #2: All comments have been addressed

2. Is the manuscript technically sound, and do the data support the conclusions?

Reviewer #1: Yes

Reviewer #2: Yes

3. Has the statistical analysis been performed appropriately and rigorously? 

Reviewer #1: Yes

Reviewer #2: Yes

4. Have the authors made all data underlying the findings in their manuscript fully available?

Reviewer #1: Yes

Reviewer #2: Yes

5. Is the manuscript presented in an intelligible fashion and written in standard English?

Reviewer #1: Yes

Reviewer #2: Yes

6. Review Comments to the Author

Reviewer #1: (No Response)

Reviewer #2: Thank you for your responses to the comments. I ask you for a minor correction to your response.

With comment 2 ("In Table 5, You did not report any p-values for linear and logistic regressions. Please add p-values to test whether, for example, high waist circumference risk significantly differs in the three indices.") I want you to perform a test to compare three indices simultaneously not pair-wise comparisons.

Notice: I have a statistician's point of view on your results. If it is not necessary to compare these three indices (PHDI, HEI, and DASH) from a nutritional point of view, please do not change Table 5.

7. PLOS authors have the option to publish the peer review history of their article (what does this mean?). If published, this will include your full peer review and any attached files.

Reviewer #1: **Yes: **Dr. Mohammad Reza Mahmoodi

Reviewer #2: No

---

## [Author Response · Author response to Decision Letter 1]

16 Nov 2023

Journal comments:

Please apply the suggestions of the respected reviewer to present the p-values for the logistic regression statistical test in cardiometabolic biomarkers between the three dietary guidelines. The consent of the statistical expert reviewer of your manuscript will be the condition of acceptance of your article.

Statistical expert reviewer comments:

With comment 2 ("In Table 5, You did not report any p-values for linear and logistic regressions. Please add p-values to test whether, for example, high waist circumference risk significantly differs in the three indices.") I want you to perform a test to compare three indices simultaneously not pair-wise comparisons.

Notice: I have a statistician's point of view on your results. If it is not necessary to compare these three indices (PHDI, HEI, and DASH) from a nutritional point of view, please do not change Table 5

Authors' reply:

We thank the reviewer and the journal for their comments and the opportunity to revise the submission. We have now jointly compared the PHDI, HEI-2015, and DASH indices using a Wald test and report a single p-value. These were done for the linear and logistic regression models whose results are reported in Table 5.

---

## [Decision Letter · Decision Letter 2]

6 Dec 2023

Dietary quality and cardiometabolic indicators in the USA: A comparison of the Planetary Health Diet Index, Healthy Eating Index-2015, and Dietary Approaches to Stop Hypertension

PONE-D-23-16525R2

Dear Dr. Frank,

We’re pleased to inform you that your manuscript has been judged scientifically suitable for publication and will be formally accepted for publication once it meets all outstanding technical requirements.

Kind regards,

Mohammad Reza Mahmoodi, Ph.D.

Academic Editor

PLOS ONE

Additional Editor Comments (optional):

Reviewers' comments:

Reviewer's Responses to Questions

**Comments to the Author**

1. If the authors have adequately addressed your comments raised in a previous round of review and you feel that this manuscript is now acceptable for publication, you may indicate that here to bypass the “Comments to the Author” section, enter your conflict of interest statement in the “Confidential to Editor” section, and submit your "Accept" recommendation.

Reviewer #2: All comments have been addressed

2. Is the manuscript technically sound, and do the data support the conclusions?

Reviewer #2: Yes

3. Has the statistical analysis been performed appropriately and rigorously? 

Reviewer #2: Yes

4. Have the authors made all data underlying the findings in their manuscript fully available?

Reviewer #2: Yes

5. Is the manuscript presented in an intelligible fashion and written in standard English?

Reviewer #2: Yes

6. Review Comments to the Author

Reviewer #2: I would like to thank to the authors for this valuable study and for their efforts to improve the results by addressing reviewers' comments.

7. PLOS authors have the option to publish the peer review history of their article (what does this mean?). If published, this will include your full peer review and any attached files.

Reviewer #2: No

---

## [Editor Report · Acceptance letter]

14 Dec 2023

PONE-D-23-16525R2 

PLOS ONE

Dear Dr. Frank, 

I'm pleased to inform you that your manuscript has been deemed suitable for publication in PLOS ONE. Congratulations! Your manuscript is now being handed over to our production team.

Kind regards, 

on behalf of

Dr. Mohammad Reza Mahmoodi 

Academic Editor

PLOS ONE